

**Refined burned-area mapping protocol using Sentinel-2 data**
**increases estimate of 2019 Indonesian burning**
David L.A. Gaveau[1] Adria Descals[2], Mohammad A. Salim[1], Douglas Sheil[3], Sean Sloan[4,5]
[1] TheTreeMap Bagadou Bas 46600 Martel, France
[2] CREAF, Centre de Recerca Ecològica i Aplicacions Forestals, E08193 Bellaterra (Cerdanyola de Vallès),
Catalonia, Spain
[3] Forest Ecology and Forest Management Group, Wageningen University & Research, PO Box 47, 6700 AA,
Wageningen, The Netherlands
[4] Department of Geography, Vancouver Island University, Nanaimo, BC, Canada &
[5]Fenner School of Environment and Society, Australia National University, Canberra, ACT, Australia
Correspondence to: David Gaveau (d.gaveau@thetreemap.com)



**Abstract**
Like many tropical forest nations, Indonesia is challenged by landscape fires. A confident understanding of the
area and distribution of burning is crucial to understanding the implications of these fires and how they might best
be reduced. Given uncertainties surrounding different burned-area estimates, and the substantial differences that
arise using different approaches, the accuracy, and merits of such estimates require formal examination.
Despite investment in fire mitigation measures since the severe El-Niño 2015 fire season, severe burning struck
Indonesia again in late 2019. Here, drawing on Sentinel-2 satellite time-series analysis, we present and validate
new 2019 burned-area estimates for Indonesia. The corresponding burned-area map is available at:
https://doi.org/10.5281/zenodo.4551243. We show that >3.11 million hectares (Mha) burned in 2019, 31% of
which on peatlands. This burned-area extent is double the Landsat-derived Official estimate of 1.64 Mha from the
Indonesian Ministry of Environment and Forestry, and 50% more that the MODIS MCD64A1 burned-area
estimate of 2.03 Mha.  It has greater reliability as these alternatives, attaining a user's accuracy of 97.9% (CI:
97.1%-98.8%) compared to 95.1% (CI: 93.5%-96.7%) and 76% (CI: 73.3%-78.7%), respectively.  It omits fewer
burned areas, particularly smaller- (<100 ha) to intermediate-sized (1000 ha) burn scars, attaining a producer's
accuracy of 75.6% (CI: 68.3%-83.0%) compared to 49.5% (CI: 42.5%-56.6%) and 53.1% (CI: 45.8%-60.5%),
respectively. The frequency–area distribution of the Sentinel-2 burn scars follows the apparent fractal-like power-
law or "pareto" pattern often reported in other extensive fire studies, suggesting good detection over several
magnitudes of scale. Our relatively accurate estimates have important implications for carbon-emission
calculations from forest and peatland fires in Indonesia. Our approach is amenable to the ongoing production of
accurate annual burned-area maps for environmental monitoring and policy in South-East Asia.

**1.    Introduction**
Landscape fires are a global concern due to their impacts. These impacts include wildlife habitat loss and
degradation, the associated emissions of greenhouse gases and toxic smoke, and the consequences for wildlife,
human health, transport, tourism, and economic activity across Southeast Asia. Fires, though scarce in wet forest
landscapes, have long been an element of traditional swidden agriculture and land clearance.  Although the causes
and motivations of modern-day fire use can be complex (Dennis et al., 2005), many fires are lit by farmers and
companies when conditions permit to burn debris and enrich the soils before planting (Gaveau et al.,
2014;Adrianto et al., 2020) or to maintain existing agricultural land (paddy fields, farm fallow). The likelihood,
scale and intensity of such fires are greatly heightened during periods of anomalously low rainfall (Sloan et al.,
2017;Field et al., 2016), as fires readily spread uncontrolled beyond the intended areas (Gaveau et al., 2017),
largely over degraded lands (Miettinen et al., 2017;Lohberger et al., 2018) but also penetrate into forest near the
edge  (Nikonovas et al., 2020). Intact rainforests don't burn without the prolonged droughts that favor the
accumulation of sufficient dry fuel, and while many live trees often remain (van Nieuwstadt and Sheil, 2005) the
resulting changes to forest structure increase the likelihood of further fires (Nikonovas et al., 2020;Cochrane,
2003). In Indonesia, droughts are often associated with years when anomalously cold sea surface temperatures
surround Indonesia and warm waters develop in the eastern Pacific Ocean (El Niño Southern Oscillation, ENSO)
and in the western Indian Ocean (Positive Indian Ocean Dipole, IOD+) (Field et al., 2009), although short, but



intense, fire episodes can also occur during climatically-normal years, or under Julian Madden weather conditions
(Gaveau et al., 2014;Koplitz et al., 2018). Austin et al. (2019) estimated that forest conversion to grasslands by
repeated fires accounted for 20% of total forest loss in Indonesia between 2001 and 2016.
The location, context, extent, and timing of fires have major implications for their impacts and their management.
During 2015, a strong El Niño year, fires burned an estimated 2.6-4.5 million hectares across Indonesia (Sipongi,
2020;Lohberger et al., 2018) and emitted 1.2 billion tons of $CO_2$ equivalent (or 884 million tons of $CO_2$) (Huijnen
et al., 2016), representing half of Indonesia's total greenhouse gas emissions for that year (Gütschow et al., 2019).
In Palangkaraya, the capital city of Central Kalimantan province, daily average particulate matter ($PM_{10}$)
concentrations often reached 1000 to 3000 µg m$^{-3}$ amongst the worst sustained air quality ever recorded worldwide
(Wooster et al., 2018). For reference, 50 µg m$^{-3}$ is a short-term (24-h) exposure limit set by the World Health
Organization (WHO), and 300 µg m$^{-3}$ is "extremely hazardous" according to by the Singapore National
Environment Agency. Over half a million people suffered respiratory problems in the aftermath, and between
12,000 and 100,000 premature deaths were estimated to result (Koplitz et al., 2016;Crippa et al., 2016). Although
2015 burning was approximately half as severe/extensive as 1997, the most severe El Niño and fire season on
record (Fanin and Werf, 2017), peatlands burned about 50% more extensively in 2015 (Fanin and Werf, 2017).
This pattern tracks a growing incidence of elevated peatland burning despite apparent long-term mitigation
(declines) to extreme fire activity (Sloan et al., Under Review).
In response to severe 2015 burning, the Indonesian government instituted several ambitious mitigation schemes.
Fire bans were enforced by dedicated command posts established in 731 fire-prone agricultural villages or *desas*
(~12 Mha), recently expanded to some 4000 village areas, with some apparent success in suppressing burning
(Sloan et al., Under Review). Simultaneously, in recognition that degraded peatlands are the primary source of
haze, the government pursued a new peatland restoration agenda. The Peatland Restoration Agency or *Badan*
*Restorasi Gambut* (BRG) was established in 2016 and declared a 2.67 Mha peatland-restoration target across 7
provinces host to >70% of the national burned area (Kalimantan Barat, Kalimantan Tengah and Kalimantan
Selatan, Papua, Jambi, Riau, and Sumatra Selatan). The seven provinces are largely the same as those actively
enforcing targeted fire bans. Restoration and fire-suppression initiatives driven by pulp-and-paper and agro-
industrial companies severely impacted by fire also flourished (Carmenta et al., 2020). These companies are
mandated to actively restore some of the targeted-for-restoration degraded peatlands (2.67 Mha).
Despite the investment in these measures since 2015, and some initial success, severe burning struck Indonesia
again in late 2019. This time a positive Indian Ocean Dipole event, rather than an ENSO weather system, was
responsible for widespread droughts, although the changing nature of these relationships and other weather
phenomenon remain a subject of ongoing research (Kurniadi et al., 2021;Cai et al., 2021). While Sloan et al.
(Under Review) suggest that 2019 fire activity was lower than might have occurred under the conditions
otherwise, the total number of MODIS active-fire detections in late 2019 was still amongst the greatest recorded
since 2001 in the village areas targeted for fire suppression, excepting 2015 (Sloan et al., Under Review).
However, counts of active-fire detections are not the same as estimates of area burned (Tansey et al., 2008) and
for 2019 such area estimates remain uncertain.
Accurate estimates of burned lands, and in particular assessments of peat fires, are key to ambitious Indonesian
climate-change atmospheric carbon (C) reduction national commitments (DGCC, 2019). Burned-area estimates





are used to calculate annual C emissions from fires, contribute to forensic analyses in landholdings (e.g. oil palm
and pulp & paper concessions), and help identify the result of policies and practices intended to reduce or control
fires, such as land enforcement and restoration of degraded lands.
Using visual interpretations of time-series Landsat-8 imagery, the Indonesian Ministry of Environment and
Forestry (MOEF) estimated that 1.64 Mha burned in 2019 (Sipongi, 2020). The commonly used global MODIS
annual burned-area product (MCD64A1, collection 6) (Giglio et al., 2018) indicated 2.01 Mha burning in 2019.
Both datasets suffer shortcomings that bias their estimates, however.  The coarse 500-m spatial resolution
MCD64A1 data omit smaller fires and thus overlook many localized events and overestimate larger ones. The
MCD64A1 dataset reports omission and commission errors of 40% and 22% globally for the 'burned' class
(Giglio et al., 2018). This validation is based on independent globally distributed visually interpreted reference
satellite data, however none over Indonesia. Conversely, the Landsat imagery underlying MOEF estimates
(hereafter 'Official estimate') are, while finer scale, observed every 16 days at best (typically much less due to
cloud and smoke), meaning that many burn scars may remain undetected.  Also, smaller-scale and/or dispersed
fire activity may be underestimated, considering the challenges of their visual interpretation and delineation.   A
thorough accuracy assessment is also not available for the official burned-area product. Given the unknown errors
around burned-area estimates, and the differences between them, the accuracy, and merits of the different mapping
approaches over Indonesia require formal examination.
Here, we present new and validated 2019 burned-area estimates for Indonesia using a time-series of the
atmospherically corrected surface reflectance multispectral images (level 2A product) taken by the Sentinel-2 A
and B satellites.  With higher spatial resolution (20-m) and more frequent observations (5-day revisit time), the
Sentinel-2A and B satellites offer relatively comprehensive and accurate burned-area mapping (Huang et al.,
2016). As detailed below, we developed our method using the Google Earth Engine (Gorelick et al., 2017), in turn
allowing for its reproduction for ongoing burned-area monitoring.  We also developed an independent reference
dataset to compare the accuracy of our estimate against the Official and MCD64A1 burned-area maps. Given the
lack of randomly distributed ground verifications of 'burned' and 'unburned' locations, we sought an efficient
way to extract reference sites by visually detecting either a smoke plume, a burn scar, or a heat source (flaming
front, or hotspot) from the archive of original time series Sentinel-2 images.  Finally, we examine differences in
terms of scar-size frequency distributions among these three burned-area estimates to examine spatial patterns.

## 2. Methods

### 2.1. Summary of methods

A burned area is an area of land characterized by deposits of char and ash, and by alteration of vegetation cover
and structure. We mapped burned areas using a change -detection approach, i.e. by comparing Sentinel-2 infrared
signals recorded before and after a burning event (Liu et al., 2020). We assembled two national composite images
depicting vegetation condition before and after 2019 burning (Figure 1) by automatically extracting pairs of
nominally 'burned' and 'unburned' pixels from 47,220 original Sentinel-2 images acquired between 01 November
2018 and 31 December 2019. This reconstructed pair of pre- and post-fire images spans the entire 2019 burning
season. It is a convenient way to capture the entire burned landscape stored in just two image files. Subsequent to
the production of this image pair, we classified pixels of the pair as 'burned' or 'unburned' using a Random Forest





classification model trained on visually-identified pairs of pre- and post-fire pixels. Third, three independent
interpreters assembled a reference dataset by visually interpretating burn scars in the original time-series (5-day
repeat pass) Sentinel-2 images. Fourth and finally, we assessed our burned-area map, as well as the Official and
MCD64A1 burned-area maps, against our reference dataset to gauge the reliability and accuracy of the three
burned-areas products.  Finally, we tested whether, and how, the three burned-area estimates differed in their
tendencies to incorporate burn scars of larger or smaller sizes.

*2.2. Pre- and post-fire Sentinel-2 national composite images of 2019*
Here, we describe our automated procedure to extract pairs of 'burned' and 'unburned' pixels from 47,220
Sentinel-2 images acquired throughout 2019. This set of pixel pairs was used to create the national composite pre-
and post-fire images and guide subsequent supervised classifications of burned areas nationally.  Prior to running
this procedure, we removed cloud-impacted pixels using the Sentinel-2 imagery quality flag (this flag provides
information about clouds, cloud shadows, and other non-valid observations) produced by the ATCOR algorithm
and included in the atmospherically-corrected surface reflectance multispectral images  of the Sentinel-2 A and B
satellites Surface Reflectance products (Level 2A product) (Fletcher, 2012).
A time series of the Normalized Burned Ratio (NBR), given as (NIR-SWIR) / (NIR+SWIR), represents a
convenient index to detect if and when a disturbance in the vegetation occurred in 2019, such as a burning
event (Key and Benson, 1999). Before a fire, vegetated pixels register high NBR values close to 1 because
reflectance in near-infrared spectrum (NIR; wavelength=0.842 µm; Band 8) is high due to the chlorophyll content
of the vegetation (open circles before fire in Figure 2). The NBR of burned vegetation typically declines due to
chlorophyl and leaf destruction, such that NBR of $\leq 0$ are apparent for a few weeks after a fire, while the
reflectance of short-wave-infrared spectrum (SWIR; wavelength = 1.610 µm or 2.190 µm; Band 11 or Band 12)
increases due to charred material and exposed ground cover. We analyzed a NBR time series for approximately
94.5 billion 400 m$^2$ pixel pairs (Indonesia's landmass =198 Mha) to detect the day when a pixel's vegetation was
disturbed by fire.
We detected breaks in NBR time series with a moving-window approach. Every two days, a moving window
scanned NBR values three months prior and one month after the central day of the window. The output value of
the moving window (blue dots in Figure 2) is the difference between average NBR values observed before and
after the central day. The day of the year when this difference reached a maximum corresponded to the moment
NBR dropped most markedly in each pixel over a two-day period, flagging a disturbance to the pixel's vegetation
potentially caused by fire. At this date, we created a pair of pre- and post-fire pixels by selecting the median Red,
NIR and SWIR spectral values acquired three months before and one month after the potential burning event. We
selected a one-month window rather than a three-month window to compute the post-fire image to maximize our
chances to detect a fresh burn scar, given that burned areas on degraded lands and savanna tend to re-green rapidly.
*2.3. Supervised burned/unburned classification.*
We used the Random Forest supervised classification algorithm (Breiman, 2001),  available via the Google Earth
Engine, to classify burned areas from the pair of pre- and post-fire image composites created above. Supervised
classifiers require 'training data', that is, exemplary spectral signatures of 'burned' and 'unburned' lands in the
present case, to guide the algorithm to reliably classify the target classes. The spectral signatures (i.e., the
reflectance values in the pre- and post-fire composite images) are the predictive variables of the classification



model. We used the NBR and all available Sentinel-2 spectral bands of the pre- and post-fire image composites
as input to the Random Forest model.
We trained the Random Forest algorithm using 988 training pixels, being point coordinates labelled as either
'burned' (317 points) or as 'unburned' (671 points). The selection of these pixels was realized by visual
interpretation of the pre- and post- fire image composites. Burned areas show a distinctive dark (low albedo)
brown/red color in the SWIR-NIR-Red composite image when displayed as Red-Green-Blue channels (Figure 1).
The training pixels were collected in a variety of landcover types to ensure the representativeness of the training
dataset and the satisfactory generalization of the classification model across Indonesia. We selected training pixels
focused explicitly on medium-to-high burn severity, i.e. areas where the distinctive red color in the SWIR-NIR-
Red composite image looked the darkest, indicating that all or most of the vegetation/soil burned.  This aspect of
the methodology hedged against over-estimation of total burned area by minimizing so-called "false positives".
It may however exclude areas with implied low-burn severity, such as understory fires (below an intact forest
canopy) and even some agricultural and grassland fires. By prioritizing confident identification of fires over
absolute burned-area coverage, as well as by duly validating our estimates, this conservative approach has the
advantage of assuaging sensitivities  concerning false positives  (Rochmyaningsih, 2020).

*2.4. Burned-area map validation.*
The Gold standard is to validate the map against a sufficiently large reference dataset developed based on ground
visits to 'burned' and 'unburned' sites sampled randomly across the country (Olofsson et al. 2014). We sought
another way to generate the reference dataset because the sample of GPS locations of 'burned' locations collected
by Indonesian government were not available. Given the laborious scale of this validation exercise, we validated
our burned-area estimates for only the seven provinces prioritized by the Indonesian Government for restoration
of fire-prone degraded lands (Kalimantan Barat, Kalimantan Tengah and Kalimantan Selatan, Papua, Jambi, Riau,
and Sumatra Selatan). These provinces are also those that typically burn most extensively.  We used visual
interpretations of the original time-series Sentinel-2 imagery acquired every 5 days over 2019 at 1298 randomly
selected sites (one site = one pixel of 20 m x 20 m) to detect flaming fronts (fire hotspots) and other signs of
burning (smoke and charred vegetation). We used these reference data to calculate the overall accuracy (OA),
producer's accuracy (PA), and user's accuracy (UA) with a 95% confidence interval, of all three burned area maps
(i.e., our Sentinel-derived burned-area classification, the official Landsat-based burned-area map, and the
MCD64A1 product) following "good practices" for estimating area and assessing accuracy reported by Olofsson
et al. 2014. We use the term '*mapped burned-area*' for the area classified as burned by each burned-area map.
We employ the term '*corrected burned-area*' for the estimation of the burned area based on the validation of a
given burned-area map against the reference dataset, following the practices in Olofsson et al. 2014. For instance,
a high omission rate in the 'burned' class of a given burned-area estimate would potentially lead to a lower *mapped*
*area* than a *corrected area* for that estimate, while a high commission rate would potentially lead to a higher
*mapped area* than the *corrected area*. The *corrected area* represents an estimation of the actual burned area for
year 2019 computed for each of the three datasets separately. The accuracy of the burned area map, and the sample
size of the reference dataset, play a role in the confidence interval of *corrected area* estimate.  Lower map accuracy
and smaller sample size mean wider confidence intervals.



### 2.4.1. *Reference site sampling design*

The good practices for estimating area and assessing accuracy reported in Olofsson et al. 2014 assume a simple random sampling or a stratified random sampling in the generation of the reference dataset. In our case study, we employed a stratified-random sampling approach to ensure an acceptable sample of 'burned' reference sites. Our stratified approach was necessary given that the 'burned' class was rare over the study area: the area of seven provinces of interest is 87.6 Mha and the combined area detected as burned by all three datasets represented only 3.1% of this area.

For the generation of the 1298 reference sites, we first randomly sampled (i) 419 sites across from the areas classified 'burned' by the three datasets (red area in Figure 3a; Supplementary Table S1), and (ii) 879 sites in areas classified as 'unburned' by all three datasets hereafter denoted U (grey area in Figure 3a). This sample size is deemed sufficient and comparable to other map assessments at larger scale (Stehman et al., 2003;Olofsson et al., 2014).

This initial sample of 1298 total sites present a shortcoming for direct pair-wise comparisons of between the reference dataset and each of the three burned-area maps individually. Specifically, sampling densities in the reference dataset were far greater in areas classified 'burned' by the three datasets (red area in Figure 3a) compared to the area deemed 'unburned' by all three datasets, hereafter denoted U (grey area in Figure 3a). Consequently, for the validation of a given burned-area dataset, its total number of 'unburned' reference sites would be over-sampled upon defining 'unburned' reference sites with reference to U as well as areas classified as burned uniquely by one of the other two maps (cyan areas in Figure 3b, c, d, hereafter denoted as U'). Such over-sampling of reference sites in the realm of U' would violate the stratified-sampling approach described in Olofsson et al. (2014) and would lead to an erroneous accuracy assessment. In order to achieve a balanced stratified sampling of reference sites across 'burned' and 'unburned' areas of each dataset, we generated three subsamples from the initial 1298 reference sites (red areas in Figures 4f,g,h) and used these subsamples to validate each dataset. These three subsamples were generated by randomly excluding reference sites from the realm of U' in Figure 3b, c and d, respectively, until the density of reference sites in U' equaled the density of the larger unburned area U. For instance, for the validation of the Official burned-area map, the density of reference sites in U was 10.36 sites/Mha, and the extent of U' was 1.551 Mha, such that the number of reference sites to retain in U' for this validation was given as 1.551 Mha x 10.36 sites/Mha = 16 sites. The calculations of the number of sites removed from each subsample are illustrated in Supplementary Table S2. The final, adjusted, stratified subsamples of reference sites used for validation is given in Table 1.



### 2.4.2. *Interpretation of the burned-area reference dataset*

We developed a series of scripts in the Google Earth Engine to streamline the visual interpretation of the reference sites. Specifically, we adapted a script written by (Olofsson et al. 2014) to rapidly scan the time-series of original Sentinel-2 images in visible and infrared bands and thus visually detect either a smoke plume, a burn scar, or a heat source (flaming front), and determine whether and when in 2019 a reference site burned. The script enabled the interpreter to interactively track the evolution of NBR values and patterns over the 2019 time series of 5-day images. Reference sites were investigated for burning wherever a marked drop in the NBR time series was detected, indicating a disturbance in the vegetation. For reference sites where a disturbed area was observed, we subsequently reviewed the last few images before the drop in NBR and the first few images after the drop. Interpreters looked for three distinct signs of burning in these images to confirm them as burned: (i) smoke plumes; (ii) flaming fronts – that is, a line a moving fire where the combustion is primarily flaming; and (iii) rapid changes in color from 'green' to 'red', characteristic of a transition to charred vegetation (Figure 4). If rapid changes in color were observed over the reference site, with at least one direct feature (smoke or flame) in its vicinity, this indicated a fresh burn scar, and the reference site was declared 'burned'. If none of these three features were observed, the reference site was declared 'unburned'.

Three interpreters independently reviewed the time-series of images and associated NBR trends for all reference sites (N=1298). To reduce uncertainties associated with the interpretation of the imagery, the results of the three interpreters were compared to each other. If all three interpreters recorded the same interpretation and timing of a burning event for a given reference site, their interpretations were retained. If one or more interpreters disagreed, all interpreters reviewed the data and resolved discrepancies by consensus. In some cases, it was difficult to reconcile disagreements because of poor image quality or because of uncertain spectral patterns. Therefore, if possible, interpreters also explored other satellite images (e.g. Landsat) to detect the presence of fire and resolve disagreements for a given reference site. The sites in which the three interpreters disagreed were ultimately excluded (70 sites) from the reference dataset. For these excluded sites, disagreement typically resulted from uncertainties over the boundary of burned or unburned areas, or because the imagery was not clear enough. The final sample size explored here, N=1298, excludes the discarded points of disagreement in question.

We created a second script to generate snapshot images (see examples in Figure 4) depicting infrared spectral conditions, shortly before and shortly after a fire, as well as the corresponding image dates. Interpreters recorded and geotagged a snapshot of before and after fire condition at every reference site (for which a burned area was detected) to enable third-party reviewers to check the consistency and validity of interpretations on site-by-site basis (See Data Availability).

### 2.4.3. *Burn scar size comparisons.*

We tested whether, and how, the three burned-area estimates differed in their tendencies to incorporate burn scars of larger or smaller sizes. Specifically, we compared the frequency distributions of burn-scar size amongst the three estimates to test for similarity and qualify any distinguishing differences on the part of our Sentinel-based estimate. Differences amongst burn-scar size frequency distributions implies that a given burned-area estimate is more or less inclusive of burn scars of a given size, regardless of absolute differences to total burned area between





the estimates. Inter-estimate comparisons of burn-scar size frequency is analogous to tests of whether the
'samples' of burn scars defined by each estimate describe the same, ultimately partially-observed universe of fire
activity. Significant inter-estimate differences imply greater or lesser inclusion of a given realm of fire activity –
e.g., small-scale agricultural burning, plantation fires, extreme wildfires – thus indicating bias (or lack thereof)
without defining such realms explicitly.
For all three estimates, we employed the Kruskal-Wallis H test of differences with respect to the 'location' of
frequency distributions along a continuum of burn-scar sizes. Given significant inter-estimate differences
according to this three-way test, we tested for two-way differences in the shape and location the scar-size
frequency distribution (Kolmogorov-Smirnov test), as well as two-way differences in medians (Mann-Whitney U
test), between our Sentinel estimate and either the Official or MODIS estimate individually. We performed all
comparisons for scar-size cohorts $> 6.25$ ha, $> 20$ ha, $> 100$ ha, $> 1000$ ha, and $> 5000$ ha, without Bonferonni
correction given the nested nature of these cohorts. Testing for similarity over increasingly large scar-size cohorts
clarified the degree to which significant inter-estimate differences were attributable to the inclusion or omission
of a given cohort.

We excluded scars $<6.25$ ha because this is the minimum observable burn scar size according to MODIS data,
given pixel resolution, and it is already evident that our Sentinel estimates are distinguished by their ability to
detect burn scars below this threshold. The Landsat-8 Official estimates similarly have few scars $< 6.25$ ha due
to the challenging nature of visual interpretations at such fine scales. In relation to Sentinel and MODIS estimates,
for which burned areas were originally mapped as arrays of pixels, we defined a burn scar to be any array of pixels
contiguous across cardinal directions but not diagonals. For the Official estimate, burn scars are as manually
delineated via visual interpretation by interpreters from the Government of Indonesia. All scars are spatially and
temporally discrete, such that scars of a given estimate that overlap spatially but not temporally are considered
separate.

## 3. Results


*3.2. Increased Burned-Area Estimates*

Our Indonesia-wide burned-area estimate, based on the classification of the pair of pre- and post-fire Sentinel-2
composites, are larger than the Official estimates as well as the MODIS MCD64A1 to a lesser degree. We estimate
3.11 million hectares (Mha) burned in 2019 across Indonesia, of which 31% were on peat (Figure 5). The extent
of peatlands were defined using a national dataset from the Ministry of Agriculture (Ritung et al., 2011). In
contrast, Official burned-area estimates, based on visual interpretation of Landsat-8 imagery, report only about
half as much burned area, at 1.64 Mha, of which 39% was on peat. Our estimates are similarly considerably
greater than the MODIS MCD64A1 product, which reports 2.04 Mha burned in 2019, or two-thirds of our
estimate, with 40% on peat. The greater burning extent and proportionally lesser extent of peatland burning
according to our estimates suggest that our estimates are particularly more inclusive of burning across mineral
soils.



In the seven provinces for which we carried out the accuracy assessment, our Sentinel-2 estimates and the Official
Landsat-8 estimates both report excellent user's accuracies (UA) for the 'burned' class, at 97.9% (CI: 97.1%-
98.8%) and 95.1% (CI: 93.5%-96.7%) respectively, indicating a mere 2.9%-4.9% commission-error rate (Table
2, Supplementary Table S3). The producer's accuracies (PA) are comparatively lower for both datasets, but
notably less so for our estimates, at 75.6% (CI: 68.3%-83.0%) and 49.5% (CI: 42.5%-56.6%) for our estimate and
the Official dataset, respectively. In other words, for any burned area in our reference dataset, there is a 75.6%
chance that it will be correctly mapped as burned by our estimate, compared to only a 49.5% for the official
estimate. This is in keeping with the greater tendency of the Sentinel-2 estimate to capture more smaller and
intermediate-size burn scars. The MCD64A1 data had a much lower UA for the burned class, at 76.0% (CI:
73.3%-78.7%), as well as a much lower and a PA for the burned class, at 53.1% (CI: 45.8%-60.5%), qualifying it
as the least reliable and accurate of the three estimates notwithstanding comparable high overall accuracy (Table
341   2).

All three burned-area maps underestimate the true burned area extent, as per their respective PA figures, but our
Sentinel-based map underestimates considerably less severely without a corresponding loss of user's accuracy.
The corrected burned area of the seven provinces is higher than the mapped area for all the three burned area
maps. Again, however, our Sentinel-based map area most closely approximates its corresponding corrected burned
area (Table 2). Whereas our Sentinel-based mapped burned area indicates that 1.84 Mha burned in the seven
provinces (or 59% of our total national estimated burned area), the corrected burned area is 2.38 Mha (CI: 2.14
Mha-2.61 Mha) (Table 2), for a discrepancy of 0.54 Mha. In contrast, the official estimate indicates 1.19 Mha
burned in the seven provinces (73% of its corresponding total), and a corrected burned area of 2.29 Mha (CI: 1.96
Mha-2.63 Mha), for a 1.1 Mha discrepancy. Likewise, the MCD64A1 dataset mapped 1.58 Mha burned in the
seven provinces and has a corrected burned area of 2.27 Mha (CI: 1.94 Mha-2.59 Mha), for a 0.69 Mha
discrepancy. Although, we cannot extrapolate a corrected burned area across Indonesia, we confidently conclude
that appreciably more than 3.11 Mha burned nationally in 2019.

354       *3.1. Burn scar size comparison.*
The Sentinel, Official and MCD64A1 estimates captured significantly distinct realms of fire activity, as
represented by their relative frequencies of scar sizes (Figure S2). The three estimates differ from one another
decreasingly over increasingly larger minimum scar-size thresholds, however, and are statistically
indistinguishable for scars > 5000 ha indicative of extreme fire activity (Table 3). In other words, all three
estimates capture very large scars (>5000 ha) equally well, and distinctions amongst the estimates concentrate
amongst small (<100 ha), intermediate (100-1000 ha) and larger (1000-5000 ha) scars, in decreasing order of
degree as indicated by the magnitude of the test statistics in Table 3.


Inclusivity of smaller and intermediate scars is the primary source of difference among estimates. Compared to
Official or MCD64A1 estimates, the Sentinel estimate has a significantly greater relative frequency of small scars
(< 100 ha), especially amongst the smallest of these scars (Table 4). This is indicative of a greater detection of
the realm of fire activity presumably characterized by small-scale agriculture fires and similar, small-scale
controlled burning. The Sentinel estimate similarly has a greater relative frequency of intermediate scars (100-
1000 ha), but less acutely so, with inter-estimate differences being more moderate for the Official estimate than



the MCD64A1 estimate (Table 4, Figure 6 Figure S2).  For scars >1000 ha, the Sentinel estimate differs only
relative to the official estimate (Table 3), seemingly due to the latter's lesser estimation of large and very large
scars (Figure 6).  Note for instance the increasingly large divergence between the cumulative burned-area curves
for the Sentinel-2 and the Official estimates in Figure 6 for scars > 1000 ha.  For very large scars (> 5000 ha),
two-way comparisons in Table 4 again report no significant statistical differences in burn-scar detection rates
between the Sentinel and alternative estimates.  However, given the small sample of patches > 5000 ha, it is
noteworthy that the Sentinel estimate captures more very large scars compared to Official estimates (n=56 vs
n=16) and avoids critical omissions made by both Official or MCD64A1 estimates for extremely large scars
(>15,000 ha) on peatlands around Berbak National Park in Jambi Province, Sumatra (Figure 1, Inset A).

In summary, the greater overall burned-area estimate of our Sentinel data compared to the Official and MCD64A1
alternatives is largely attributable to differences in the inclusion of smaller and intermediately sized scars.  Indeed,
the aerial sum of all Sentinel burn scars that are individually <~860 ha equals the entirety of the official burned-
area estimate (Figure 6).  While the finer spatial resolution of Sentinel data must account for some of the inter-
estimate discrepancies, particularly relative to the MCD64A1 estimate and scars < 100 ha (Figure S2), overall the
discrepancies above seem more in keeping with our estimate's greater sensitivity to otherwise overlooked smaller-
scale burning.  Hence, the inter-estimate differences qualify our Sentinel estimates not simply as more extensive
but also as qualitatively distinct in terms of the degree to which different realms of fire activity are captured. The
near linear log-log frequency–area distribution over several orders of scar-size of our Sentinel product indicates a
characteristic power-law relationship (Figure 6).
**4. Discussion**
We developed a method that generates two national composite Sentinel-2 images depicting vegetation condition
before and after burning in 2019 (Figure 1), and then classified this pair to extract burned areas using a Random
Forest supervised classification algorithm. We developed a comprehensive validation protocol to strictly assess
the reliability and accuracy of our product based on visual interpretation of dense time-series Sentinel-2 original
images, and also applied this validation to the  widely used global MODIS burned-area product (MCD64A1,
collection 6) (Giglio et al., 2018) and to the Official burned-area product of the Indonesian Ministry of
Environment and Forestry (MOEF) (Sipongi, 2020).
Our estimate is the most reliable and accurate and therefore captures more of the 2019 total burned area,
confirming that 20-m Sentinel-2 imagery  is better suited to widespread small-scale agricultural burning  in
Indonesia (Huang et al., 2016), while it also captures large burn scars relatively thoroughly. The study finds similar
omission and commission errors (47% and 24%) for the 'burned' class of MCD64A1 product as those presented
globally (40% and 22%) (Giglio et al., 2018). The underestimation of total burned area according to the
MCD64A1 product compared with our Sentinel-2 estimate is unsurprising, considering that the MODIS 500-m
pixel resolution struggles to detect smaller fires (Giglio et al., 2018).  More surprising is the near 2:1 ratio by
which the Sentinel-2 estimates surpass the Landsat-8 Official estimate.  Our examination shows that this
difference reflects differential detection of small- (<100 ha) to intermediate-sized (<1000 ha) burn scars.
The burn-scar frequency distribution of the Sentinel-2 estimate is characteristic of robust power-law relation
(Figure 6), a pattern typical of large scale fire studies (Malamud et al., 1998). Modern studies suggest that these
fractal-like patterns are often subtly more complex and can arise through a range of phenomena  (Karsai et al.,



2020;Falk et al., 2007). We note that the Sentinel-2 estimate exhibits a size-frequency pattern that approximates
the linear expectation of a near scale-free power-law, or pareto distribution, compared to either of the alternative
burned-area estimates, both of which show a clearly S-shaped curve with less area at smaller and larger sizes,
indicating the bias by omission. These results, with different frequency patterns arising from burns from the same
regions in the same period, also highlight the danger in interpreting apparent burned-area patterns without careful
consideration of the limitations and biases that arise from the methods used to map them—an issue that may not
have always been sufficiently recognized in past assessments or policy.
Although both Sentinel-2 and Landsat-8 both observe the infrared wavelengths required to detect charred
vegetation and have similar spatial resolutions (20 m x 20 m and 30 m x 30 m, respectively), Sentinel-2 detects
more burns of the greater frequency of its coverage (five- versus sixteen-day revisit time).  Also, our method
avails of the massive computational capabilities and automation of the Google Earth Engine, allowing us to
analyze more images and thus map more and smaller burn scars and associated details than could even the most
well-equipped team of visual interpreters.
Despite high reliability that every burn scar detected on the map was valid (2.9% commission error rate), our
method suffered a 24.4% omission error rate (burned areas that remained undetected).  These rates reflect
necessary tradeoffs between commission and omission error in a context where conservative estimates are much
preferred for environmental policy and monitoring.  We prioritized a low commission error rate (i.e. high user's
accuracy) over absolute burned-area coverage to address sensitivities (Rochmyaningsih, 2020). By hedging
against commission errors, our approach omitted hard-to-detect events, including low-intensity burns, such as
those that occur beneath the forest canopy on mineral soils (van Nieuwstadt and Sheil, 2005) or on savanna
grasslands, which tend to re-green rapidly. While further work is required to clarify and refine the optimal levels
of inclusivity and reliability, we emphasize that the production of before and after fire annual composite images
is relatively straightforward for the user community, given the availability of both the necessary imagery and our
Google Earth Scripts.
Sometimes commentators raise doubts about our ability to confidently estimate burn scars without extensive and
costly on-the-ground ground-truthing. Modern high-resolution remote sensing makes such on-the-ground checks
less essential than in the past as burned areas are readily identified with good accuracy in modern high-resolution
imagery such as that we used for our validation. The protocol developed here to generate a reference dataset based
on visual inspection of dense (5-day revisit time) satellite imagery is better suited than ground verifications of
'burned' and 'unburned' locations, because it allows the generation of extensive randomly-distributed well
characterised reference sites, a process too time-consuming and costly with field visits. The identification and
quantification of less-readily-detected burned areas, such as those under a closed forest canopy, remain a challenge
but will require dedicated and targeted research and would not be solved by ground-checks alone.
Accurate estimates of burned lands, in particular on peat, are central to address concerns about regional air quality,
and to ambitious national climate-change atmospheric carbon reduction commitments heavily reliant on improved
land/fire management (DGCC, 2019). Though we observed proportionally less peatland burning than the
alternative burned-area estimates (31% versus 39% and 40% for the Official and MCD64A1 products,
respectively), due to our more complete coverage, we observed more peatland burning absolutely (0.96 Mha) than
the official estimate (0.64 Mha). Given this large discrepancy for peatland burning, we anticipate that our
improved mapping approach will become a "gold-standard" reference to calculate carbon emissions from the 2019



fires in Indonesia. Combined with daily fire hotspots detected using thermal remote sensing, our detailed burned-
area map can help identify ignition sites and estimate fire duration more precisely, and therefore contribute to
forensic analyses of burning across landholdings (e.g. concession owners) as well as assess policies and practices
intended to reduce or control ignition events and the scale of fires (Watts et al., 2019).
The Indonesian government has shown some success in reducing fires (Sloan et al., in review). Apparent
reductions to fire activity would however ideally be qualified using our more inclusive and accurate burned-area
estimates. Further, the Indonesian government must also develop improved protocols to quantify the resulting
carbon emissions (DGCC, 2019). Our protocols for creating reliable and accurate burned area maps are replicable.
To further the adoption and reproduction of our approach, we have published all our protocols, scripts,
applications, burned-area map, reference data, pre-fire and post-fire Sentinel-2 composite images, and various
other outputs so that anyone may employ and revise them as they wish (see Data Availability).

### 5. Code availability

The code that generates the Sentinel-2 pre- and post-fire composites can be found at:
https://github.com/thetreemap/IDN_annual_burned_area_detection

### 6. Data Availability

All the data including pre- and post-fire composites, all three burned area products, and reference points with
screenshots can be visualized online at this application portal:
https://thetreemap.users.earthengine.app/view/burn-area-validation-simplified
The Sentinel-based burned area map and reference dataset are freely available for download at:
https://doi.org/10.5281/zenodo.4551243.
The dataset *2019_burnedarea_indonesia.shp* contains the 2019 burned-area estimates that we developed for
Indonesia using 20 m x 20 m time-series Sentinel-2 imagery. The reference dataset *Reference_dataset.shp*
contains 1298 reference points that we assembled and used to validate all three burned area products described in
this study. Each reference point includes attribute 'REFERENCE' to describe the values obtained by visual
interpretation: either 'NO' unburned or 'YES' burned. Each reference point has three attributes: 'C_SENTINEL'
'C_OFFICIAL' and 'C_MCD64A1' to describe the values of the classification of each burned area product: either
'NO' unburned or 'YES' burned. Finally, each reference point has three additional attributes: 'SENTINEL',
'OFFICIAL', and MCD64A1' to describe which burned area product this reference point validates. The values
are either 0: not validate or 1: validate.
The MODIS MCD64A1 dataset was obtained at: https://developers.google.com/earth-
engine/datasets/catalog/MODIS_006_MCD64A1. The official burned area dataset from the Ministry of
Environment and Forestry (MOEF) was obtained at: https://geoportal.menlhk.go.id/webgis/index.php/en/
The Sentinel-2 Level 2A used in this study are available at https://scihub.copernicus.eu/ and can be retrieved in
Google Earth Engine. The Sentinel- 2 data are hosted and accessed in the Earth Engine data catalog (the links to
the data are https://developers.google.com/earth-engine/datasets/catalog/COPERNICUS_S2_SR). Data ingested
and hosted in Google Earth Engine are always maintained in their original projection, resolution, and bit depth
(Gorelick et al., 2017).




**Financial support.** Funding by the CGIAR Research Program on Forests, Trees and Agroforestry (CRP-FTA),
with financial support from the donors to the CGIAR Fund, is recognized.

**Author Contributions.** D.L.A.G. designed the study. DL.A.G, M.A.S. and A.D designed the burn scar detection
method. M.A.S. and A.D wrote the code in Google Earth Engine. D.L.A.G, M.A.S. and A.D. carried out the
validation. S.S. carried out the burn scar size analysis. D.L.A.G., A.D. S.S. and D.S. interpreted the results and
wrote the manuscript and produced the figures.

**Competing interests.** The authors declare no competing interests. Readers are welcome to comment on the online
version of the paper.

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

**Figures**



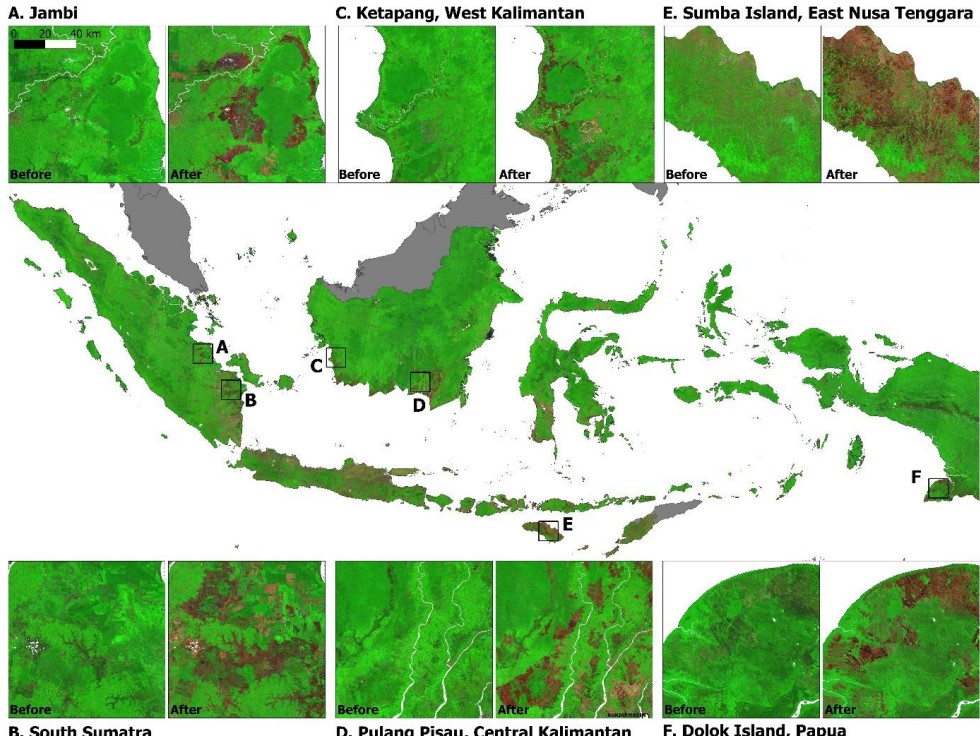

**Figure 1. T**he pair of cloud-free pre-and post-fire Sentinel-2 composites shown over six locations in insets A, B, C, D, E, F (all insets have the same scale). The base Indonesia-wide imagery is the post-fire composite. Imagery displayed in false colors (RGB: short-wave infrared (band 11); Near infrared (band 8), Blue: red (band 4)). In this pair of composite images acquired shortly before and after fire a recently burned area will readily appear to have transitioned from 'green' to dark 'brown/red' tones. Areas cleared without burning appear bright pink. Areas covered with vegetation appear dark to bright green.



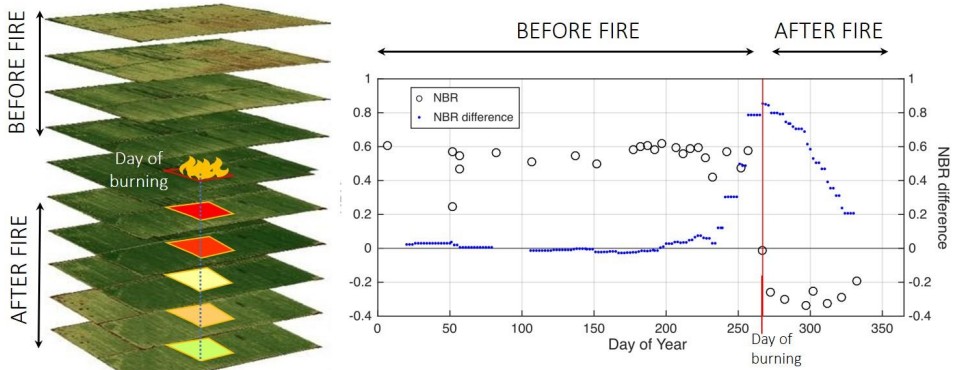

**Figure 2.** A schematic of Sentinel-2 time-series imagery, associated NBR values (open circles) and NBR differences between average NBR values observed before and after the central day of a 2-day moving window (blue dots). A burned pixel (20 m x 20 m) is represented by a red rectangle at left. Before fire, the vegetated pixel registers positive NBR values (open circles). The NBR rapidly drops during the fire and, for a few weeks, the satellite observations show a negative NBR. The day of the year when the NBR difference observed via the moving window reaches a maximum corresponds to the moment NBR dropped (red line). This day marks a decline in the pixel's vegetation, possibly reflecting a burning event. Over time, the vegetation regenerates (re-greening) and the spectral characteristic of charred vegetation fades. Regreening can happen within days in the case of savanna grasslands, or within months in the case of forest fires on peatlands.

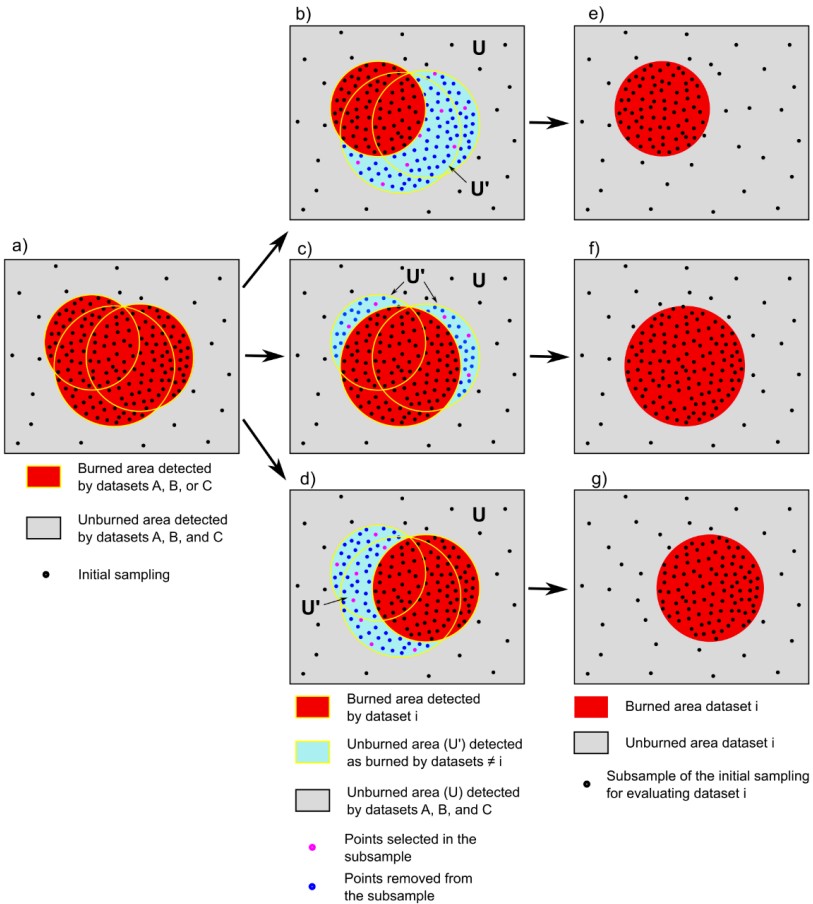

**Figure 3.** Representation of the adjusted, stratified-sampling design for the validation of three burned area datasets (A, B, and C) against reference sites (dots). Panel (a) shows the stratified random sampling of reference sites (black points) over the combined burned area. Note that the density of samples is higher in the combined burned area than the unburned area. Panels (b), (c), and (d) show, in cyan, the area U', being classified as unburned in a given dataset $i$ but classified as burned in at least one other datasets $\neq i$. For a given validation of A, B, and C, the sample points in the corresponding area U' (panels (b), (c), (e)) were randomly excluded until the sampling density in the area U' equaled that of the larger unburned area U (area in gray). Panels (f), (g) and (h) show the three final, adjusted, stratified subsamples of reference points derived from the initial sample of 1298 reference points. Note that the relative areas and number of sites per class in Figure 3 do not correspond to the actual datasets being evaluated.

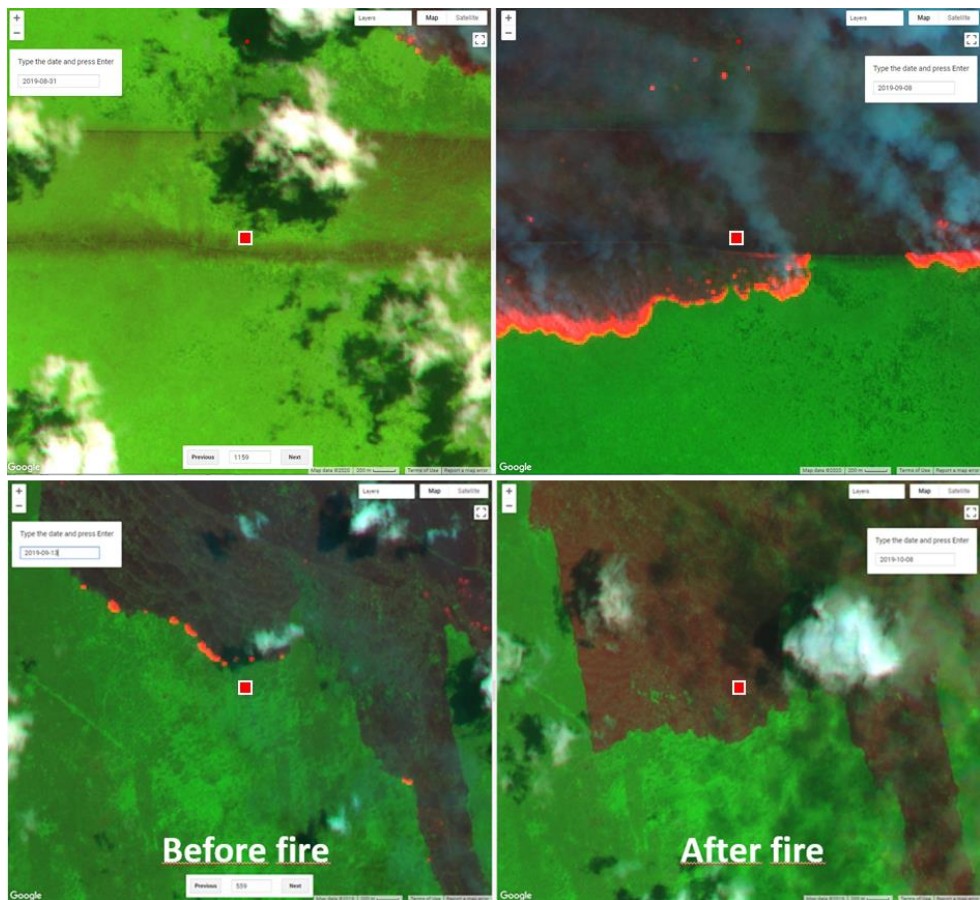

682

**Figure 4.** Two snapshots recording the pre-fire (left panel) and post-fire (right panel) original Sentinel-2 images acquired shortly before (13 September 2019) and shortly after (08 October 2019) fire for two reference site (red squares). Imagery displayed in RGB: SWIR, NIR, RED. Sentinel-2 provides two SWIR Bands. Band 12=2.190 µm is more suitable than Band 11=1.610 µm to detect the intense heat from flaming fronts. On these image pairs, one can see flaming fronts traveling towards the reference sites (red dot) from the north on the pre fire images (left), and sharp changes in color from 'green' to 'dark red' characteristic of charred remains with continuing flaming on the post-fire images (right). Layout built using © Google Earth Engine.






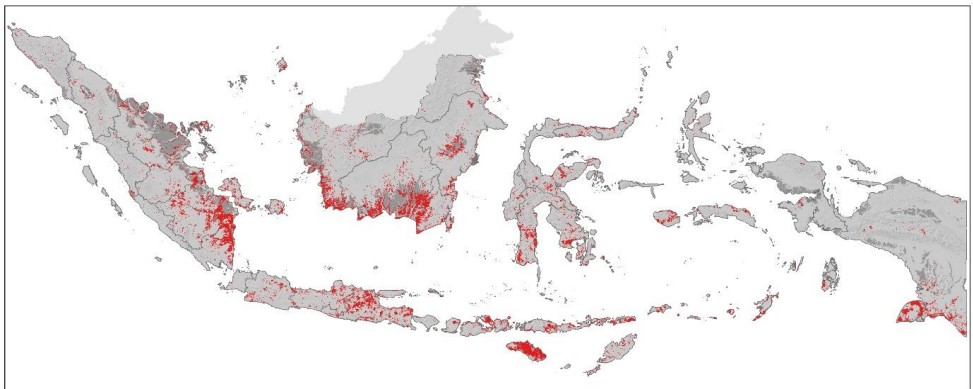


**Figure 5.** 2019 burned areas (red) for Indonesia derived using a time-series of the atmospherically corrected surface reflectance multispectral images (level 2A product) taken by the Sentinel-2 A and B satellites. The spatial resolution of this map is 20 m x 20 m, and Minimum mapping unit is 6.25 ha. The officially recognized peatlands extent is shown with the darkest shade of grey. A provincial breakdown of burned areas according to our map estimates and those of the Official and the MCD64A1 product are given in Figure S1.

699

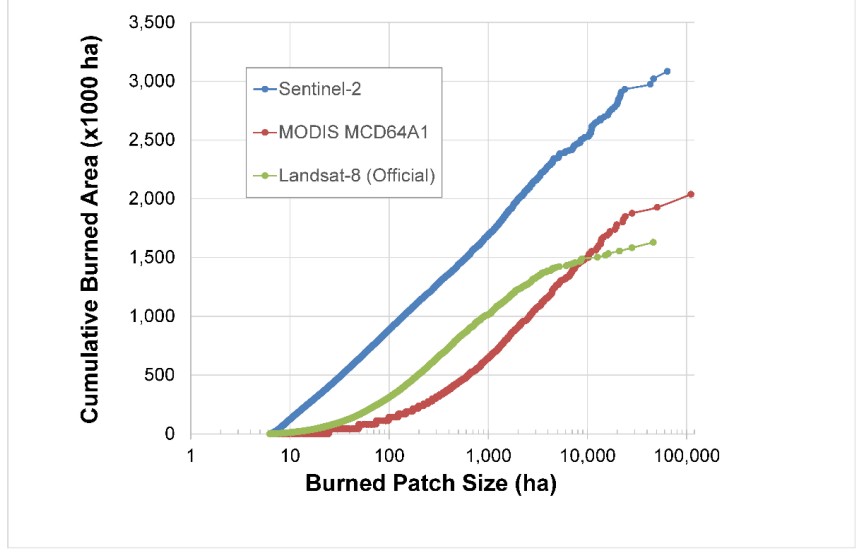

700

**Figure 6.** Cumulative national total burned area versus burned-scar area, for Sentinel-2, Landsat-8 (Official), and MODIS MCD64a1 burned-area estimates. Scars < 6.25 ha are not shown. Note the logarithmic axis. For a given segment of the x-axis between scar sizes $X_1$ and $X_2$, a difference in the slopes for any two estimates is indicative of inter-estimate differences in terms of inclusivity of scars between $X_1$ and $X_2$.

705

706

707





**Tables**

**Table 1.** Adjusted, Stratified Subsamples of Reference Sites to Validate Burned-Area Estimates.

| **Burned-Area Estimate** | **Reference Sites** | | **Total Reference Sites** |
| --- | --- | --- | --- |
| | *In Areas Classified as Burned* | *In Areas Classified as Unburned (U & U')* | |
| Sentinel-2 (this study) | 888 | 280 | 1168 |
| MODIS MCD64A1 | 891 | 242 | 1133 |
| Landsat-8 (Official) | 895 | 182 | 1077 |

**Table 2.** Accuracy assessment of each of the three burned area maps performed in seven Indonesian provinces (87.60 Mha) targeted for peatland restoration. The accuracy metrics were estimated with an initial total of 1,298 points randomly distributed using stratified sampling. The reported metrics are: 1) the overall accuracy (OA), the user's accuracy (UA), and the producer's accuracy (PA) with their 95% confidence intervals, and 2) the mapped burned area and the corrected burned area with their 95% confidence intervals.

| | | *SENTINEL* | *OFFICIAL* | *MCD64A1* |
| --- | --- | --- | --- | --- |
| *OA (%)* | | 99.3 (99.1, 99.6) | 98.1 (97.8, 98.5) | 98.4 (98.1, 98.8) |
| *UA (%)* | *Burned* | 97.9 (97.1, 98.8) | 95.1 (93.5, 96.7) | 76.0 (73.3, 78.7) |
| | *Unburned* | 99.3 (99.1, 99.6) | 98.6 (98.2, 99.0) | 98.8 (98.5, 99.2) |
| *PA (%)* | *Burned* | 75.6 (68.3, 83.0) | 49.5 (42.5, 56.6) | 53.1 (45.8, 60.5) |
| | *Unburned* | 99.9 (99.9, 99.9) | 99.9 (99.9, 99.9) | 99.6 (99.6, 99.7) |
| *Mapped burned area (Mha)* | | 1.84 | 1.19 | 1.58 |
| *Corrected burned area (Mha)* | | 2.38 (2.14 , 2.61) | 2.29 (1.96 , 2.63) | 2.27 (1.94 , 2.59) |
| *Difference (Mha)* | | 0.54 | 1.1 | 0.69 |

**Table 3.** Tests statistics with respect to three-way differences in burned area scar-size frequency distributions for Sentinel, MODIS, and official estimates.

| **Scar Size (ha)** | **Kruskal-Wallis H[a]** |
| --- | --- |
| **> 6.25** | 10,478** |
| **> 20** | 998* |
| **> 100** | 335* |
| **> 1000** | 14* |
| **> 5000[a]** | 0.61 |

Significance: ** p<0.0001; * p<0.001
Notes: Scar-size thresholds in the table denote the set of scars included in a test. Tests pertain to whether frequency distributions have equivalent 'distribution location', that is, position along a continuum of scar sizes. Tests thus pertain to whether the estimates capture distinct realms of fire activity, assuming similarly shaped frequency distributions. Higher test statistic values indicate greater probability that the estimates differ with respect to distribution location. The tree-way comparisons of the estimates may flag differences where all three estimates differ or where only two of the three differ. Significance is not Bonferroni corrrected. (a) There are 56, 60 and 16 scars > 5000 ha for Sentinel, MCD64A1, Official estimates, respectively.





**Table 4.** Test statistics with respect to two-way differences in burned area scar-size frequency distributions, with respect to distribution shape and situation (Test I) or situation alone (Test II), for Sentinel estimates compared to either MCD64A1 or Official estimates.

| Scar Size (ha) | Sentinel vs. MCD64A1 | | | Sentinel vs. Official | | |
|---|---|---|---|---|---|---|
| | *I. Kolmogorov-Smirnov Z-score (Most Extreme Difference [positive/negative])[b]* | | *II. Mann-Whitney U Z-score* | *I. Kolmogorov-Smirnov Z-score (Most Extreme Difference [positive/negative])[b]* | | *II. Mann-Whitney U Z-score* |
| > 6.25 | 46.9** (+0.69) | | -82.9** | 31.8** (+0.32) | | -70.6** |
| > 20 | 14.7** (+0.24/0.-15) | | -20.1* | 13.2** (+0.18) | | -28.6* |
| > 100 | 7.9** (+0.23) | | -16.6* | 1.6† (+0.04/-0.04) | | -0.57 |
| > 1000 | 0.76 (+0.06/-0.03) | | -0.79 | 1.5‡ (+0.01/-0.12) | | -3.1• |
| > 5000[a] | 0.72 (+0.14/-0.08) | | -0.77 | 0.70 (+0.13/-0.20) | | 0.10 |

Significance: ** p<0.0001; * p<0.001; • p<0.01; † p=0.014; ‡ p<0.05

Notes: Scar-size thresholds denote the cohort of scars included in a test. Test I and Test II both pertain to whether the Sentinel estimates capture distinct realms (scar-size cohorts) of fire activity compared to the other two estimates. Test I pertains to whether the scar-size frequency distribution of the Sentinel estimate has the same shape and 'distribution location' as either the MODIS or official estimate. Test II is the same but with respect to distribution location only. Distribution location refers to the situation of a frequency distribution along a continuum of scar sizes. Higher test statistics indicate greater probability that the estimates differ significantly with respect to distribution shape and/or location. Reported statistical significance is without Bonferroni corrections. a) There are 56, 60 and 16 scars > 5000 ha for Sentinel, MODIS, official estimates, respectively. (b) Largest positive and negative differences in the cumulative probability functions of Sentinel vs. MODIS or official scar-size estimates. No difference was reported where it was <0.00 absolutely.