# Peer review of "Refined burned-area mapping protocol using Sentinel-2 data"

_Earth System Science Data, 2021_

## Author Response (AR1)

We are grateful for the many suggestions and comments. These have helped us revise the manuscript both in terms of the specific issues raised, and in terms of helping us reorganise and clarify our emphasis and give the article greater coherence. We have made many larger and smaller changes in the revision that improve the manuscript. Here we describe specific responses to specific queries (below) made by three referees, but we underline that the entire article has been revised for flow, clarity and focus.

**Referee 1**

Overall, this is an interesting topic and highlights the various limitations in burned area products. I applaud the effort that went into creating the validation dataset, however, as highlighted below, there are some crucial details lacking in the training and validation descriptions that are necessary for a full review. Please see below for specific comments.

We are grateful for the astute comments raised by Referee #1 as they have helped us clarify and improve the manuscript. Here, we answer against each point below.

**Specific/Major Comments**

Section 2.1: I assume the November and December 2018 images were only used for the pre-fire window analysis? How did the authors account for any burns that occurred during those 2 months?

The images taken in November and December 2018 were only used for the pre-fire compositing. Fires that occurred during these two months were not detected. We re-wrote sections 2.1 and 2.2 to improve clarity.

Section 2.2: What happens if a pixel burns more than once in a year? This might occur in the agricultural regions?

Our detection method does not record multiple burning events in a year. The day of the year when the NBR difference dNBR reached a maximum corresponded to the moment NBR dropped most markedly in each pixel, flagging a disturbance to the pixel's vegetation potentially caused by fire. We reworded section 2.2 to this point clearer.

Line 162: Are the authors not calculating dNBR here? Also, does the moving window continue beyond the first instance of detecting a potential burn? i.e. if the drop occurred on February 1st, does the moving window continue to Dec 31 to see if it burned again?

The moving window calculated dNBR values throughout the year and retained the date with the highest dNBR, flagging a disturbance to the pixel's vegetation potentially caused by fire. At this date, we created a pair of pre- and post-fire pixels by selecting the median Red, NIR and SWIR spectral values acquired three months before and one month after the disturbance. We repeated this procedure for approximately 94.5 billion pixels to assemble two national composite images depicting the spectral condition of vegetation shortly before and shortly after a disturbance (Figure 1). After the production of the pre- and post-fire composites, we used a Random Forest classification model (see section 2.3) trained on visually identified pairs of pre- and post-fire pixels to confirm if the spectral changes indicating vegetation damage corresponded to a burning event.

If a fire occurred on 01 February, and the resulting dNBR recorded on this day was the maximum difference all the differences calculated (every two days) throughout the year, this day was retained as the day of the year when a disturbance to the pixel's vegetation was potentially caused by fire. If the resulting dNBR was not the maximum difference, this day (01 February) was not retained. That

would be the case if another more severe fire occurred at the same pixel later in the year, which would result in a higher dNBR.

We reworded sections 2.1 and 2.2 to improve clarity.

Line 177: Can the authors please provide a map of the locations of the 988 training pixels and their associated land cover types in the supplementary? Is 988 training pixels enough? How was that number decided upon?

We have included Supplementary Figure S1 to show the location of the 988 training points used to train our supervised classification algorithm (Random Forest).

[Figure]

**Figure S1.** Location of 988 training pixels (317 'burned' and 671 'unburned') used to train our supervised classification model (Random Forest) across Indonesia (grey area).

The required number of points used to train our supervised classification model (here a Random Forest) depends on the spectral separability of the classes (in our case two classes: "class burn" and "class not-burn"). The pixels that show the burn scar present a singular spectral signature and, for this reason, it is necessary to collect a particularly high amount of training points. We collected training points until we were satisfied with the results of the classification by visual inspection. Please note that the training points differ from the validation points. They do not overlap. An adequate validation set is important as the number of validation points limits how narrow the confidence intervals are.

We added supplementary Table S3 to show that the training pixels were collected in a variety of landcover types.

**Table S3.** Landcover types associated with the training sites one year before fire (2018) based on the ESA CCI global land cover maps described here (http://maps.elie.ucl.ac.be/CCI/viewer/index.php). The training sites were associated with 15 landcover types.

| CCI 2018 Land Cover | Unburned | Burned |
|---|---|---|
| Cropland, rainfed | 54 | 33 |
| Herbaceous cover | 58 | 44 |
| Tree or shrub cover | 29 | 6 |
| Cropland, irrigated or post-flooding | 6 | 0 |
| Mosaic cropland (>50%) / natural vegetation (tree, shrub, herbaceous cover) | 93 | 52 |
| Mosaic natural vegetation (tree, shrub, herbaceous cover) (>50%) / cropland | 40 | 28 |
| Tree cover, broadleaved, evergreen, closed to open (>15%) | 132 | 47 |
| Mosaic tree and shrub (>50%) / herbaceous cover (<50%) | 0 | 5 |
| Mosaic herbaceous cover (>50%) / tree and shrub (<50%) | 2 | 12 |
| Shrubland | 9 | 0 |
| Sparse vegetation (tree, shrub, herbaceous cover) (<15%) | 31 | 39 |
| Tree cover, flooded, fresh or brakish water | 57 | 33 |
| Tree cover, flooded, saline water | 18 | 0 |
| Urban areas | 136 | 18 |
| Water bodies | 6 | 0 |
|  |  |  |
| Total number of training sites | 671 | 317 |

Line 183: dNBR already shows burn severity (here is an article with more information: https://un-spider.org/advisory-support/recommended-practices/recommended-practiceburn-severity/in-detail/normalized-burn-ratio). Did the authors quantify these values over their training pixels or simply rely on the color? I suggest the authors quantify these values to ensure the training pixels are in fact medium-to-high severity especially since the authors are prioritizing mapping high burn severity fires to reduce false positives.

Thank you for providing the link. We assessed burn severity during training based on visual interpretation. RGB composites with bands 11 (SWIR wavelength = 1.610 µm), 8 (NIR wavelength=0.842 µm) and 4 (RED wavelength = 0.665 µm) provide information about the severity of the fire; burn scars with high severity present a dark (low albedo) red/brown color. We understand that visual interpretation can be subjective. We included the histogram of dNBR ($NBR_{postfire}$ - $NBR_{prefire}$) for the 317 training points labelled 'burned' in Supplementary Figure S2 to corroborate that the training samples were selected in areas with medium to high severity fires.

81% (256) of 'burned' training points (317) had dNBR values ($NBR_{postfire}$-$NBR_{prefire}$) < - 0.44, which represents the threshold for medium to high severity burns according to the proposed classification table of the United States Geological Survey (USGS).

**Training data**
**dNBR Burned area: mean=-0.61632 std=0.18443**
$n_{dNBR<-0.44}=256$ $n_{dNBR>-0.44}=61$

Figure S2. Histogram of dNBR ($NBR_{postfire}$ - $NBR_{prefire}$) for the 317 training points labelled 'burned'.

Section 2.4.1: As with the training samples, what land cover types were associated with the validation samples?

We added a Supplementary Table S4 to show the landcover types associated with the reference sites one year before fire (2018) based on the land cover maps described here (http://maps.elie.ucl.ac.be/CCI/viewer/index.php ). The Reference sites were associated with 14 landcover types. Here the 'burned' and 'unburned' classes are the 'truth' labels deemed 'burned' by visual inspections.

**Table S4.** Landcover types associated with the reference sites one year before fire (2018) based on the ESA CCI global land cover maps described here (http://maps.elie.ucl.ac.be/CCI/viewer/index.php ).

| CCI 2018 Land Cover | Unburned | Burned |
|---|---|---|
| Cropland, rainfed | 16 | 13 |
| Herbaceous cover | 16 | 3 |
| Tree or shrub cover | 23 | 6 |
| Cropland, irrigated or post-flooding | 1 | 0 |
| Mosaic cropland (>50%) / natural vegetation (tree, shrub, herbaceous cover) | 127 | 101 |
| Mosaic natural vegetation (tree, shrub, herbaceous cover) (>50%) / cropland | 150 | 73 |
| Tree cover, broadleaved, evergreen, closed to open (>15%) | 467 | 63 |
| Mosaic tree and shrub (>50%) / herbaceous cover (<50%) | 5 | 6 |
| Mosaic herbaceous cover (>50%) / tree and shrub (<50%) | 7 | 20 |
| Sparse vegetation (tree, shrub, herbaceous cover) (<15%) | 16 | 33 |
| Tree cover, flooded, fresh or brakish water | 94 | 31 |
| Tree cover, flooded, saline water | 19 | 4 |
| Urban areas | 1 | 0 |
| Water bodies | 3 | 0 |
|  |  |  |
| Total number of reference sites | 945 | 353 |

Secondly, what size burn scars were these validation pixels associated with? For example, if all validation pixels were associated with very large burn scars then the validation results will be biased because large burns are easy to detect. Also, I assume the training and validation samples did not overlap?

Our reference sites (i.e., validation pixels') were associated with a wide range of burn sizes. The uppermost histogram in the Figure below (added as Supplementary Figure S3) shows the frequency distribution of Sentinel-2 burn scar sizes for scars coincident with a subset of our 1298 reference sites used to validate our Sentinel burned-area map and deemed 'burned' by visual inspections. As seen, the patches coincident with these reference sites range from very small (a few hectares) to very large (over 60,000 ha). This diversity of patch size is to be expected considering that reference-site sampling was realised randomly across the entirety of burned areas, without regard to patch size, with the partial exception that patches <6.25 ha were excluded from consideration for reasons noted in the main text. Correspondingly, the positive skew of the reference-site histogram is in keeping with the positively skewed frequency distribution of all Sentinel burned-area patches for Indonesia, shown in the lower histogram of the Figure below.

Notwithstanding the points above, the frequency distributions of the upper and lower histograms in Figure are ultimately statistically different from one another, insofar as the distribution for the reference sites is *comparatively* biased towards larger patches. The most likely reason for the statistical difference in question is that the hyperabundance of very small patches in our Sentinel-2 burned-area map would require an exceptionally large sample of reference sites to fully represent such small patches alongside a proportional diversity of intermediate and larger patch sizes.

[Figure]

[Figure]

**Figure S3.** Frequency distributions of patch sizes of the Sentinel burned-area map, for select spatially coincident reference sites used to validate the Sentinel burned-area map (top), and for all of Indonesia (bottom). Note: Bin widths are not consistent between upper and lower panels. In the lower panel, note the logarithmic scale of the y-axis and the presence of rare patches above 40,000 ha. Patches <6.25 ha are excluded. Reference sites are those 274 sites deemed 'burned' by visual inspections (labelled as 'truth') and coincident with Sentinel-2 burns.

Finally, we affirm that our training and validation samples were independent and did not overlap.

Line 310-311: Please explain why the authors only chose the cardinal directions?

The choice to define burned-area patches in terms of pixel contiguity in the cardinal directions but not on diagonals was intended to render the resultant burned-area map conservative with respect to patch size. Given that the Sentinel data has a relatively fine spatial resolution (10 m), minor sub-pixel burning within a single 'burned' pixel could conceivably link two much larger burned areas into a single discrete patch if diagonal contiguity were recognized. Such an outcome would, in our view, potentially inflate overall patch-size estimates, perhaps especially for smaller-scale patches for which burning often adopts 'patchy' spatial patterns. Of course, the same issue also applies to a single pixel contiguous in a cardinal direction. However, our preliminary inspections of the geography of burning in our Sentinel burned-area map suggested that 'undo' contiguity along individual diagonal pixels was more common and/or potentially problematic. We revised the text to justify our choice accordingly.

Line 389: While doing some reading into power laws and fire size, I came across this paper from the US Forest Service with the following quote: "*Newman (2005) specifically excludes fire size distributions, while admitting that they might follow power laws over portions of their ranges. Current opinion is divided among those who would globally assign power laws to fire-size distributions (Minnich 1983; Bak et al. 1990; Malamud et al. 1998, 2005; Turcotte et al. 2002; Ricotta 2003) and those who would attribute them only to portions of distributions or rule them out altogether in favor of alternatives (Cumming 2001; Reed and McKelvey 2002; Clauset et al. 2007; Moritz et al., Chap. 3)*" - https://www.fs.fed.us/rm/pubs_other/rmrs_2011_mckenzie_d001.pdf

Please can the authors go through the literature and ensure their power-law assumption is correct and justify it in the paper.

We are not making any theoretical claims or assumptions that fire size distribution follows a power law. We just note that this pattern has been observed by other studies, and is observed over a greater range of scales in our refined burned area analysis as we would expect if these methods are better able to detect burns (which is our main point here, and why this emphasis is helpful). Indeed, as we note in the discussion the comparisons also highlight how the detection of these patterns depends on the nature of the methods used to detect them, which is something that is not appreciated in the published literature around this theory. We revised the text to clarify our focus.

Line 399: The current analysis does not support this finding regarding agricultural burning. Based on Figure S3, the small patches are likely associated with the small burn patches surrounding the larger burn scars. Agricultural burning is a very difficult fire type to map and although the current methodology is likely to map more agricultural burning than MODIS (due to the finer resolution) the mapping methods and validation assessment was not adequately designed for agricultural burning. The authors can mention that the S2 mapping is better suited for identifying "small fires". Furthermore, it was noted on lines 427 – 433 that the approach omitted hard–to–detect fires (e.g. savanna grasslands) which are much easier to detect than agricultural fires therefore that statement is not supported.

We rephrased accordingly by removing the word 'agricultural': "Our estimate is the most reliable and accurate and therefore captures more of the 2019 total burned area, 399 confirming that 20-m Sentinel-2 imagery is better suited to widespread small-scale  burning in Indonesia"

Minor Comments

Line 23: change to "..which occur on.."

We rephrased accordingly.

Line 27: Should the size of the intermediate fires read (100ha – 1000ha) similar to what you have in the main body of the paper?

Thank you for noting this error. We have corrected it.

Line 88: change "excepting" to "except"

Thanks, changed

Introduction: When are the peak burning months? It seems based on the GWIS country profiles (https://gwis.jrc.ec.europa.eu/apps/country.profile/charts ) that the peak occurs August – October and since the authors are also referring to agricultural fires then please also include the cropland burning months.

We have added that: *Most fires occur during drier months (July to October) and the threats are greatly heightened during years of anomalously low rainfall.*

Line 137: Remove "and finally" after "Fourth. The authors go on to a final step on line 139.

Fixed

Figure 3: There is no h panel (line 679)

Thank-you for spotting this mistake. We fixed it

Line 237: Should that be referencing Figure 3? Also, there is no panel h

Thank-you for spotting this mistake. It should read Figures 3e,f,g

Line 696: change to "minimum mapping unit"

Thanks. Fixed

Figure 5: Please add to the caption that the light grey represents countries outside of Indonesia. This was confusing at first before I looked at a map.

We removed the grey areas outside of Indonesia to improve the clarity of Figure 5. We also removed the outline of 'burned' polygons during the display of the burned area shapefile used to generate the map because the outline display exaggerated the burned area extent visually. The new Figure 5 is below.

[Figure]

**Figure 5.** 2019 burned areas (red) for Indonesia (grey) derived using a time-series of the atmospherically corrected surface reflectance multispectral images (level 2A product) taken by the Sentinel-2 A and B satellites. The spatial resolution of this map is 20 m x 20 m, and minimum mapping unit is 6.25 ha. The officially recognized peatlands extent is shown with the darkest shade of grey. A provincial breakdown of burned areas according to our map estimates and those of the Official and the MCD64A1 product are given in Figure S1.

Table S3: please add the meanings of Am and Wh to the caption

We added meanings of Am and Wh in Figure caption

**Table S3.** Confusion matrix. Am = Area mapped (the area classified as class i by the Random Forest; the sum of this column is equal to the total area of study). Wh = Proportion of area mapped (the proportion of area classified as class i; the sum of this column equals to 1)

Line 370: add a comma between "Figure 6 Figure S2"

Fixed

Line 702: Change to "MCD64A1". There were a few other instances where the A was lowercase (i.e. MCD64a1)

Fixed

Line 377: It would be interesting for the authors to create a 3-panel figure showing this scar from each of the 3 products to show the omissions made in MCD64 and the Official dataset.

We added a 6-panel Figure in the main text (Figure 7) to illustrate this case.

[Figure]

**Figure 7.** The pair of cloud-free pre-and post-fire Sentinel-2 composites over Berback National Park (black line) and surrounding areas in Jambi Province (see also Inset A, Figure 1), revealing large, burned areas around Berbak National Park (areas that have transitioned from 'green' to dark 'brown/red' tones). These large burn scars have been detected by VIIRS hotspots and by the Sentinel-2 burned area map, but some have been missed by the Official and MCD64A1 datasets.

Line 443: change to "addressing"

Fixed

**Reviewer2**

The MS has a clear structure and a good description of reference data collection. However, my major concern is the design of the mapping framework. The most confusing part is the integration of NBR and RF classification. As stated by the authors, NBR and dNBR were used to "detect the day when a pixel's vegetation was disturbed by fire". If this is valid, what was the point of using RF for burned area mapping? On the other hand, RF can be directly used for classification. What was the benefit of combining NBR and RF? Another concern is that the authors put the research in the context of national burned area mapping in Indonesia only, but there is a lack of clarification on the scientific contribution of the work. There is no literature review of the state-of-art on large-scale burned area mapping. There is no scientific objective (what scientific issue was addressed), and no comparisons with similar studies on the topic (expect for the comparison with the Official and the MODIS results).

We are grateful for the comments raised by Referee #2 as they have helped us clarify the aim of this study, methodology, and improve the manuscript. We rewrote the introduction to shift the emphasis to burned area mapping. We feel that our methodology was sometimes misunderstood, and we hope to have clarified our methods.

Thank you for pointing out that literature review of the state-of-art on large-scale burned area mapping was missing. We have added a review in the Introduction which is now extensively rewritten and reorganised. As stated by Reviewer 3, a focus on Indonesian burning is important, given the significant impacts that severe burning episodes in Indonesia have on the global carbon cycle and on human health across Southeast Asia. A key aspect of our study is the comparison of our burned area dataset against two other studies (the official study and the MCD64A1, which is considered one of the most accurate global burned area product) using a rigorous validation procedure. The refined burned area product presented in this study represents an important development and will be of interest to the scientific community because accurate estimates of burned lands, and in particular assessments of peat fires, are key to better measure atmospheric carbon emissions. Our goal in this paper in line with the aim of this Journal, which is to publish articles on original research datasets, furthering the reuse of high-quality data of benefit to Earth system sciences.

Regarding methods (the use of NBR and RF), we revised sections 2.1, 2.2 and 2.3 (see manuscript with track changes) to describe the benefit of combining NBR and RF. Our revision went through several rounds among the authors to ensure it was clear to each of us.

dNBR time-series, despite its name (Normalized Burned Ratio) cannot determine whether fire alone is the cause of damage. For example, tree cutting exhibit a similar drop in NBR as burned vegetation. The NBR difference (dNBR) was used to detect the day when a pixel's vegetation was potentially disturbed by fire, but it could be another cause, for example a cutting event (e.g. mechanical conversion to agriculture, to timber plantation, to roads, to population centers, mining or natural timber harvesting), a disease, strong winds, floods, or landslides. We refer to the satellite composites as "pre- and post-fire composites". However, the pre- and post- fire composites capture all types of vegetation damage that may have occurred throughout the year (e.g fire, tree cutting, landslide, disease, etc..). Tree cutting exhibit a similar drop in NBR as burned vegetation, but the spectral changes before and after the disturbance are different. Usually, the albedo is lower in burned vegetation than in clear cut areas, and this difference is best captured using a carefully trained Random Forest. Thus, the NBR time-series was used to create two national composite images depicting the spectral condition of vegetation before and after a damaging event (potentially a fire), while the Random Forest was used to determine if vegetation damage was caused by fire. The features used in the Random Forest are the bands of Sentinel-2 in the pre- and post-fire composites plus their respective NBR index. We excluded the bands at 60-meter spatial resolution (bands B1, B9, and B10) since these bands present a low spatial resolution for the aim of the study. Therefore, we used a total of 22 features; the NBR and bands B2, B3, B4, B5, B6, B7, B8, B8A, B11, and B12 of the pre and post-composites.

The classification of pre- and post-fire composites represent a more effective way to capture the changes in the time series than using directly the RF for the classification of single satellite images for two reason. First, pre- and post-fire composites depict land cover changes while single satellite images just show specific instants in the time series. Although both post-fire composites and single satellite images show burn events, the pre-fire composite provides useful information that might improve the accuracy of the RF classification. For instance, the change burned-to-burned would not be classified as burned in the pre- and post-fire composite approach because the RF is trained such as an area that has previously burned cannot burn again. Second, in terms of computing time, the classification of two images, pre- and post-fire composites, is arguably a more convenient approach than the classification

of 47,220 original image files used to create them. The reviewer can visualize this pair of composites at this application portal against the classified results: https://thetreemap.users.earthengine.app/view/burn-area-validation-simplified.

Other comments follow:

Introduction: The background info about wildfire is too long, especially for the first four paragraphs. The emphasis should be burned area mapping.

We agree, thank-you for the suggestion. We rewrote this to shift the emphasis. We added a literature review of current state-of-the-art global and regional mapping products with several new references added. (Alonso-Canas and Chuvieco, 2015; Lizundia-Loiola et al., 2020; Otón et al., 2019; Chuvieco et al., 2019, Hawbaker et al., 2017, Lohberger et al., 2018, Ramo et al., 2021).

L108: Isn't visual interpretation more accurate than machines? Many field data are from visual interpretation, including yours. It is important to point out the issues in their visual interpretation strategy/method/data.

We inserted this sentence in the introduction: *Visual interpretation entails a manual delineation of burn scars perimeters, which yields accurate results for large burn scar mapping at local scales, but is too time consuming at large spatial scales, particularly when mapping small fires*.

L109: It is surprising that such a national campaign does not have protocols for accuracy assessment.

It is indeed the case that accuracy assessment is not available for the official burned-area product. Our study fills this gap.

Section 2.2.: The purpose of getting the pre- and post-fire composites with NBR is confusing. Why did you use RF classification since you already identified the burned pixels with NBR? Or why not directly using RF to extract burned areas?

We hope to have clarified this point earlier. We reworded sections 2.1, 2.2 and 2.3 (see manuscript with track changes) to describe the benefit of combining NBR and RF. Our revision went through several rounds among the authors to ensure it was clear to each of us.

L163-165: The description of implementing your method is vague. What NBR and dNBR thresholds did you use? How did you know the variation of NBR was caused by wildfire, not other events (e.g., plant disease)? Do you also need to have a vegetation baseline map?

Again, we hope to have clarified this point earlier. The NBR time series (see Figure 2) was used to detect the day when vegetation was damaged whether fire or another cause. Despite its name, the Normalized **Burned** Ratio (NBR) and the Normalized Burned Ratio difference (dNBR) cannot determine the cause of vegetation damage. For example, tree cutting events exhibit a similar drop in NBR as burned vegetation, but the spectral changes are different. Usually, the Albedo is lower in burned vegetation than in clear cut areas, and this difference is best captured using a carefully trained Random Forest. We used a Random Forest to determine if vegetation damage was caused by fire. We did not need to use any vegetation baseline map. The revision makes this clearer.

Section 2.3: How did you tune RF? What parameters did you use?

We added the following text in section 2.3.

*The features used in the Random Forest are the bands of Sentinel-2 in the pre- and post-fire composites plus their respective NBR index. We excluded the bands at 60-meter spatial resolution (bands B1, B9, and B10) since these bands present a low spatial resolution for the aim of the study. Therefore, we used a total of 22 features; the NBR and bands B2, B3, B4, B5, B6, B7, B8, B8A, B11, and B12 of the pre and post-composites.*

*We used a 10-fold cross-validation in order to assess the accuracy obtained with a set of different parameters in the Random Forest. The splitting 'train-test' in the cross-validation was done only with the training dataset, since the reference dataset used for the final validation must be completely independent of the training and model parametrization. The two parameters that we tuned were the number of trees and the minimum leaf size (Figure R1). We found that a minimum leaf size equal to 1 performs the best on average and, thus, we used this value. For a minimum leaf size equal to 1, the overall accuracy saturated for the accuracy plateaued for a number of trees >25, however we selected a conservative number of trees, 50, in order to ensure the good performance of the RF. Please note that the only trade-off when using a larger number of trees is that the RF requires more processing time.*

We did not set any limit to the maximum nodes in each tree and the variable to split in the random forest was set to the square root of the number of variables, which is the common practice among machine learning practitioners and also the default configuration in Google Earth Engine.

**L177: Aren't the training samples too small for national scale mapping?**

The required number of training points depend on the spectral separability of the classes (in our case two classes: "class burn" and "class not-burn"). For instance, the classification of water bodies is less complex than the classification of forest types (deciduous versus coniferous trees). This is because water has a distinctive spectral signature, low reflectance in most of the light spectrum, which makes its detection relatively easy; a low number of training samples can detect water bodies and more samples would only add redundancy to the classification. For our case study, vegetation-to-burn changes also show a distinctive spectral signature that can only be confused with clear cuttings (vegetation-to-bare soil). This is why a relatively small number of points (988) can accurately detect the burned areas nation-wide, and it is not necessary to collect a particularly high amount of training points. We collected training points until we were satisfied with the results of the classification by visual inspection of our pre- and post fire composites. We note that the training points differ from the validation points. They do not overlap.

We have included Supplementary Figure S1 to show the location of the 988 training points used to train our supervised classification algorithm (Random Forest).

[Figure]

**Figure S1.** Location of 988 training pixels (317 'burned' and 671 'unburned') used to train our supervised classification model (Random Forest) across Indonesia (grey area).

Results: You compared your overall results with the Official and the MODIS results. However, it is also important to pick sample locations to demonstrate what types of areas had high agreements and what areas caused discrepancies.

It is noteworthy that the Sentinel estimate captures more very large scars compared to Official estimates (n=56 vs n=16) and avoids critical omissions made by both Official or MCD64A1 estimates for extremely large scars (>15,000 ha). These omissions occur particularly on peatlands, for example around Berbak National Park in Jambi Province, Sumatra (Figure 7). Section 3.1. explains the differences in burn scar size between the three datasets. We have also added a new Figure (Figure 7) to show that the Sentinel Estimate avoids critical omissions made by both Official or MCD64A1 estimates for extremely large scars (>15,000 ha).

[Figure]

**Figure 7.** The pair of cloud-free pre-and post-fire Sentinel-2 composites over Berbak National Park (black line) and surrounding areas in Jambi Province (see also Inset A, Figure 1), revealing large, burned areas around Berbak National Park (areas that have transitioned from 'green' to dark 'brown/red' tones). These large burn scars have been detected by VIIRS hotspots and by the Sentinel-2 burned area map, but some have been missed by the Official and MCD64A1 datasets.

**Reviewer 3**

The study presents a new burned area product for the year 2019 in Indonesia based on high spatial resolution Sentinel 2 imagery and machine learning classification algorithm. Given the significant impacts that severe burning episodes in Indonesia have on global carbon cycle and population health across the wider region, the product presented in this study represents an important development and will be of interest to the scientific community. The approach and the dataset, nonetheless, have several limitations which I believe should be better articulated in the revised manuscript. In addition, I don't think that the comparison of fire patch size distributions between different products adds much to the discussion here due to (i) large differences in spatial and temporal resolutions (or both) between the datasets and (ii) lack of definition what does fire patch represents here.

While the validation methodology does seem robust and the authors do demonstrate that total burned area estimates of the study are more accurate when compared to the alternative sources (MCD64A1 and the Official ba product), it has to be articulated that the algorithm of this study was optimised for the specific region and fire season and for a specific commission/omission error ratio. As a result, it is not clear how the burned area estimate for 2019 would change if the algorithm was optimized to fit training data from different years and regions by different users. In addition, extending temporal coverage of the dataset is not that straightforward as this would require substantial further work (somewhat arbitrary and time-intense selection of training data). Please see the bellow specific comments for further detail.

We are grateful for the comments raised by Referee #3. We expanded on the limitations of our methodology, in particular when it comes to applying the algorithm to other years or other regions by adding a paragraph in Discussion. The point we are making regarding the comparison of patch size distribution is simply to highlight that a more complete size spectrum is likely indicating that we missed less burned areas, and that these patterns are influenced by size-dependent detection bias also in many other such cases that assume the nature of spectra indicates something about the fires (when it may simply reflect methods)

**Specific comments**

Lines 56-57: Given the uncertainty in burned area estimates (line 56), the Huijnen et al., 2016 estimate of CO2 emissions quoted in line 57 seems too certain. Do Huijnen et al give uncertainty estimate? Also, would be good to give another estimate for the event, given by GFED or Lohberger et al., (2018) or some other study etc. Large uncertainties in emission estimates is yet another reason why we need better burned area products, hence it would be good to point this out here.

We rephrased this sentence to incorporate the various estimates reported by several studies, including the ones reviewer 3 recommended. We wrote: *fires emitted between 0.89 and 1.5 billion tons of $CO_2$ equivalent (Huijnen et al., 2016; Lohberger et al., 2018; Van Der Werf et al., 2017)*

The GFED estimate of 1.5 *billion tons of $CO_2$ equivalent* is reported by *Van Der Werf et al., 2017*

Lines 160: Please explain what "Every two days" means here.

We have rewritten the text to better explain this: "*The difference between the average NBR values was estimated every 2 days in the time series, skipping the day of year that was an odd number (day of year equal to 2, 4, 6, 8...).*"

Line 161: Was data from the central day of the window included in prior or after median values (or neither)?

The central day of the window was included in the after median values. We have added this information in the main text: "*The NBR average after the central day also included the value of central day*"

Line 163: This relates to the previous two points regarding temporal precision. Here and elsewhere the authors use "The day of the year". How day of burn was determined if temporal resolution of Sentinel 2 is ~5 days as stated earlier? This suggests considerable uncertainty in day of burn estimate?

Thank you for pointing this out. The text was unclear about the temporal precision. Indeed there is an uncertainty in the burn date estimate. We revised any statement claiming that the exact date of burn was estimated with the NBR time series. Moreover, we added the following text in the Section 2.2.: "Although the Sentinel-2 has a temporal resolution of 5 days, the overlap between satellite passes may increase the temporal resolution regionally up to 2 days in the equator. Thus, we estimated the NBR difference every 2 days instead of 5 days. Taking this into consideration, our burn date estimate has a maximum temporal precision of 2 days in specific regions, but generally 5 days when satellite passes do not overlap."

Lines 173-176: It is not clear here what was the total number of features used for classification? Please state in this paragraph.

Thank you for pointing this out; this information was incomplete in the text. Since we did not apply any feature selection technique, the total number of features is the original bands of Sentinel-2 in the pre- and post-composites plus their respective NBR index. We excluded the bands at 60-meter spatial resolution (bands B1, B9, and B10) since these bands are mostly designed for atmospheric correction and present a low spatial resolution for the aim of the study. Therefore, we used a total of 22 features; the NBR and bands B2, B3, B4, B5, B6, B7, B8, B8A, B11, and B12 of the pre and post-composites. We included this information in the mentioned paragraph.

Lines 177-178: Was the training sample fully independent from the validation sample? This is important to state clearly as it underpins the validity of the study's findings.

The training samples are fully independent from the validation samples. They do not overlap. We have included Supplementary Figure S1 to show the location of the 988 training points used to train our supervised classification algorithm (Random Forest).

[Figure]

**Figure S1.** Location of 988 training pixels (317 'burned' and 671 'unburned') used to train our supervised classification model (Random Forest) across Indonesia (grey area).

Lines 260-265: What was done with classification of the sites which had either direct fire evidence (flame or smoke) but not indirect evidence (reddening) observed and vice versa? Where these samples (if any) discarded from the analysis?

We have modified the text to take into account these possibilities. We wrote: *If rapid changes in color were observed over the reference site, with at least one direct feature (smoke or flame) in its vicinity, this indicated a fresh burn scar, and the reference site was declared 'burned'. If rapid changes in color from 'green' to 'dark red' were observed without smoke or flame, the reference site was also declared 'burned'. If no change in color was observed, with at least one direct feature (smoke or flame) in its vicinity, the reference site was declared 'unburned'. If none of these three features were observed, the reference site was declared 'unburned'.*

Line 278: The final validation sample number N=1298 is the same as given in line 269 (all reviewed sites) and line 227 where it is termed as "initial sample". Please clarify this.

We removed the word "final" to remove any confusion. The final, adjusted, stratified subsamples of reference sites used for validation of the three burned area datasets is given in Table 1.

Line 306: This statement needs a reference and an explanation. MODIS burned area pixel size is ~21ha. While sub-pixel burning can be detected, the actual minimum burned scar size will depend on environment/vegetation where burning is occurring. Is this estimate of 6.25ha is specific to Indonesia/tropical regions? In addition, I am puzzled by how MCD64A1 fire size histogram shown in Fig. S2 was computed; i.e. how counts for bins for fires < 21ha were derived given that MODIS pixel size is ~21ha?

We thank you for spotting this error. Given the 500-m grid size of the MCD64A1, 500-m * 500-m = 25 ha is the minimum size, not 6.25 ha.

The apparent existence of MCD64A1 burn scars < 25 ha, upon re-checking our data we realised that such scars reflected a minor data-processing oversight. Specifically, it sometimes occurred that a given MODIS burn scar was 'split' or 'clipped' into two scars, one being > 25 ha (as per the original scar) and another < 21 (a so-called scar 'sliver'). This occurred exclusively where a MODIS BA patch was intersected by GIS data defining the administrative boundaries of Indonesia, i.e., along borders, coastlines, and broad river courses. Those scar 'slivers' of < 25 ha amounted to only 0.044% of the total burned area estimated by the MODIS data, and were by any measure inconsequential. Upon omitting these slivers from consideration, we have precluded any confusion over the minimum scale of the MODIS burn scars, while our methods and results have remained unchanged. The absence of MODIS scars < 25 ha is now apparent in the revised Figure 6, Table 3, Table 4, and Figure S6.

Regarding the 6.25 ha threshold. We excluded scars <6.25 ha in the Sentinel-2 product because this is the minimum observable burn scar size of the Landsat-8 Official estimates due to the challenging nature of visual interpretations at such fine scales.

Lines 366-367: The sentence is too "wordy" and complex. "greater detection of the realm of fire activity characterized by small-scale…" could be replaced with "greater detection of small fires" to the same effect.

Yes thank you. We reworded accordingly:

Line 371: "lesser estimation" – perhaps change to underestimation?

Yes thank you. We reworded accordingly

Lines 380-389: The paragraph is too wordy. The first two sentences say nearly everything that needs to be said. Sentinel 2 sensor can indeed detect smaller fires enabling the detection of small scale agricultural burning. Perhaps cut shorter or even merge into previous paragraphs.

Yes, we revised the text to make it shorter.

Lines 407-416: This paragraph is very speculative and not well supported. Not sure I agree with such interpretation of fire size frequency distributions. Any differences in distribution shape may arise from huge differences in sensor spatial (MODIS) and temporal (Official map) resolutions and also from the clustering (patch agglomeration) method. For example, the algorithm of the official product may have merged diagonally adjacent pixels as well and that would result in shift towards larger sizes. In addition, the provided references do not show that power-law approximates fire event sizes in Indonesia. While I'm not aware of fire size studies in Indonesia, log normal fire size distributions are common in some ecosystems (see Lehsten et al., 2014). As a result, it is perhaps better to avoid saying that fire sizes should follow power-law relationship and that this itself is a desirable property.

Yes, much more could be said, but we decided that that would be tangential to the main focus. The point is simply to highlight that a more complete size spectrum is likely indicating that we missed less … and that these patterns are influenced by size-dependent detection bias also in many other such cases that assume (incorrectly) the nature of spectra indicates something about the fires (when it may simply reflect methods). We have revised this paragraph to ensure that we are not claiming any broader theoretical insights (that is a distraction here) but using it to (just) help illustrate and enrich the comparisons.

Lines 423-433: While advantages of the approach are discussed across several paragraphs in the Discussion, this is the only paragraph considering the limitations. Please add discussion on implications of changes in training dataset (due to different selection criteria, addition of data from different year etc.) on burned area estimates for 2019 and beyond in future application of the algorithm.

Thank you for raising this important caveat. We have included the following paragraph in the Discussion section to discuss the limitations of our training dataset:

*While the accuracy assessment proved that our training dataset is valid for the classification of Sentinel-2 composites for the year 2019 in Indonesia, this training dataset might not achieve equivalent accuracy for other years and regions. The pre- and post-fire composites might show different spectral changes under different conditions. For instance, high rainfall in 2020 influenced reflectance. Similarly, representative training points should be used in other regions. Those adapting these methods should ensure adequate local training data and validation.*

Lines 434-435: Not clear who are those "commentators" and "us" in "our ability" the authors refer to in the sentence. Please be more specific.

Revised to be specific and the problem words are no longer present.

Lines 448-449: I do not understand why "large discrepancy for peatland burning" between the datasets would make the dataset of this study a "gold-standard"? Please explain this bold statement. Also,

please consider replacing "gold-standard" with something less flashy as only time will tell how the dataset fares among users.

We removed the word "gold-standard" and revised the text using more neutral terminology

---

## Author Response (AR2)

Thank you for addressing my previous comments. The updated manuscript has been greatly improved. Please see minor comments below.

We are grateful for the many suggestions and comments. Here we answer against each point below

Minor Comments

Line 9: Remove "&" at the end

Fixed

Line 70: The FireCCILT10 (Beta product) was recently replaced by the FireCCILT11 (Otón et al., 2021)

Otón, G., Lizundia-Loiola, J., Pettinari, M. L., & Chuvieco, E. (2021). Development of a consistent global long-term burned area product (1982–2018) based on AVHRR-LTDR data. International Journal of Applied Earth Observation and Geoinformation, 103, 102473.

Thank you for pointing this new reference. We replaced the old reference with the new one

Line 72: change to "products"

Fixed

Line 78: Hawbaker released an updated Landsat BA product over CONUS

Hawbaker, T. J., Vanderhoof, M. K., Schmidt, G. L., Beal, Y. J., Picotte, J. J., Takacs, J. D., ... & Dwyer, J. L. (2020). The Landsat Burned Area algorithm and products for the conterminous United States. Remote Sensing of Environment, 244, 111801.

Thank you for pointing this new reference. We replaced the old reference with the new one

Line 83: change "The MOEF" to "the MOEF"

Fixed

Line 85: Does the MCD64A1 product specifically omit small fires or simply not capture them all because of the coarse resolution?

We modified the sentence to make this point clearer

Line 137: I believe it should be spelled "chlorophyll"

Fixed

Line 180: capitalize random forest

Fixed

Line 197: add a ")" after 2005

Fixed

Line 210 (also Lines 438 – 446): An alternative method to ground samples is to validate a BA product using a finer-resolution sensor (e.g. validate a Sentinel 2 BA product using Planet (3m) data)

Yes thank you for pointing this out. Planet will be another possibility in future endeavours.

Line 273: change to "line of moving fire"

Fixed

Line 306-307: Please reword the sentence.

We reworded the sentence to improve clarity

Line 323: change to "burns are manually"

Changed

Line 349: Should the "and a" be removed in "…lower and a PA.."

Yes, fixed

Line 376 and 386 and 452: Capitalize "official"

Fixed

Line 417: add "because" between "burns of"

Yes fixed

Line 741: change to "reference sites"

Changed

Line 744: change to "pre-fire images"

Changed